# A Theoretical and Experimental Approach to the Analysis of Hydrogen Generation and Thermodynamic Behavior in an In Situ Heavy Oil Upgrading Process Using Oil-Based Nanofluids

Oscar E. Medina [1,2,*], Santiago Céspedes [1], Richard D. Zabala [2], Carlos A. Franco [2], Agustín F. Pérez-Cadenas [3], Francisco Carrasco-Marín [3], Sergio H. Lopera [4], Farid B. Cortés [1,*] and Camilo A. Franco [1,*]

[1] Grupo de Investigación en Fenómenos de Superficie—Michael Polanyi, Departamento de Procesos y Energía, Facultad de Minas, Universidad Nacional de Colombia, Sede Medellín, Medellín 050034, Colombia
[2] Vicepresidencia Técnica de Desarrollo (VDE), Ecopetrol S.A., Bogotá 110311, Colombia
[3] Grupo de Investigación en Materiales de Carbón, Departamento de Química Inorgánica, Facultad de Ciencias—Unidad de Excelencia Química Aplicada a Biomedicina y Medioambiente, University of Granada (UEQ-UGR), 18071 Granada, Spain
[4] Grupo de Investigación en Yacimientos de Hidrocaburos, Departamento de Procesos y Energía, Facultad de Minas, Universidad Nacional de Colombia, Sede Medellín, Medellín 050034, Colombia
* Correspondence: oemedinae@unal.edu.co (O.E.M.); fbcortes@unal.edu.co (F.B.C.); caafrancoar@unal.edu.co (C.A.F.)

**Abstract:** This study aims to show a theoretical and experimental approach to the analysis of hydrogen generation and its thermodynamic behavior in an in situ upgrading process of heavy crude oil using nanotechnology. Two nanoparticles of different chemical natures (ceria and alumina) were evaluated in asphaltene adsorption/decomposition under a steam atmosphere. Then, a nanofluid containing 500 mg·L$^{-1}$ of the best-performing nanoparticles on a light hydrocarbon was formulated and injected in a dispersed form in the steam stream during steam injection recovery tests of two Colombian heavy crude oils (HO1 and HO2). The nanoparticles increased the oil recovery by 27% and 39% for HO1 and HO2 regarding the steam injection. The oil recovery at the end of the displacement test was 85% and 91% for HO1 and HO2, respectively. The recovered crude oil showed an increment in API° gravity from 12.4° and 12.1° to 18.5° and 29.2° for HO1 and HO2, respectively. Other properties, such as viscosity and content of asphaltenes and resins with high molecular weight, were positively modified in both crude oils. The fugacity of $H_2$ was determined between the reservoir and overburden pressure and different temperatures, which were determined by the thermal profiles in the displacement test. The fugacity was calculated using the application of virial equations of state with mixing rules based on the possible intermolecular interactions between the components. Hydrogen acquired a higher chemical potential via nanoparticle presence. However, the difference in $H_2$ fugacity between both points is much higher with nanoparticles, which means that hydrogen presents a lower tendency to migrate by diffusion to the high-pressure point. The difference between HO1 and HO2 lies mainly in the fact that the pressure difference between the reservoir and the overburden pressure is greater in HO2; therefore, the difference in fugacity is greater when the pressure differential is greater.

**Keywords:** hydrogen; fugacity; crude oil upgrading; nanoparticles; steam injection

## 1. Introduction

In recent years, the challenges of energy supply and global warming have drawn increasing attention from humanity [1]. The world is facing a severe problem known as the greenhouse effect. The consequences of this phenomenon are an increase in the Earth's average temperature of 0.2 °C per decade and an increase in the concentration of $CO_2$ in the atmosphere, which can cause disastrous and irreversible changes to our planet's ecosystem [2].

Hydrogen is an essential energy vector for decarbonization, which allows the development of a clean, sustainable energy source with a low carbon footprint. In addition, hydrogen has important implications for other factors, such as reducing greenhouse gas emissions at the end-use point, enhancing the security of energy supply, and improving economic competitiveness, among others, and it is also considered a potential fuel for the transport sector [3]. However, most of the technologies used to produce gray and blue hydrogen involve releasing large quantities of $CO_2$ [4]. Moreover, these processes use large additional amounts of fossil fuels as an energy source and are highly endothermic. On the other hand, hydrogen production from renewable sources is still disadvantageous economically [5].

Nowadays, there is a need to promote technologies that help us transition from fossil fuels to sustainable energy systems such as hydrogen. Therefore, during the energy transition, it is important to establish processes that help conventional technologies to improve their energy efficiency by minimizing the amount of $CO_2$ emitted into the atmosphere and the environmental impacts [6]. An exciting strategy to foster energy transition is the co-production of hydrogen and fossil fuels with a low carbon footprint [7]. Some processes, including steam and air gasification, have been widely applied in coal, natural gas, and light hydrocarbons [8,9]. However, the use of heavy (HO) and extra heavy crude (EHO) oils for this purpose appears as a novel research topic accompanied by several challenges and considerations [10–14].

Around 70% of the worldwide reserves are from HO and EHO, representing significant economic value [15]. There are substantial difficulties associated with the high viscosity and content of asphaltenes and resins of high molecular weights [10,16–18].

Commonly, to improve the mobility and production of HO and EHO, thermal treatments are used in situ [18–30]. Most conventional processes inject steam in different ways, such as continuous steam injection [19,20], cyclic steam injection [20,21], and steam-assisted gravity drainage (SAGD) [18,22]. However, these techniques are limited by different mechanisms, including high operation costs [23], steam condensation [24], and temporal oil viscosity reduction with no change in crude oil quality [25], obtaining recovery factors close to 50% and low calorific gaseous products, including greenhouse gases (GHG) such as $CO_2$ [26–31].

The efficiency of the steam injection could be enhanced by adding chemical additives and solvent-based chemicals, mainly light to medium hydrocarbons, which can reduce steam requirement, heat losses, and GHG emissions, and increase HO productivity [32]. According to the Canadian Energy Research Institute, steam injection and SAGD generate 60.4 kgCO$_2$eq/bbl, which can be reduced between 15–20% using steam solvent and 10–15% with steam-chemical additives [33]. However, several limitations, such as low thermal stability, high costs, and low possibility to upgrade the HO, are associated with chemical additive usage.

Consequently, nanoparticles have been extensively explored in the field of heavy oil recovery, assisting conventional thermal treatments such as steam injection. A clear understanding of how nanoparticles interact with crude oil is an area of extensive research. Authors have made tremendous efforts to understand parameters such as the rheology of heavy oil, as well as compositional changes to obtain insights on how crude oil upgrading can be achieved [18,34–38]. As mentioned before, heavy oils are laden with asphaltenes in the bulk, which imparts them with their semi-solid structure. Breaking—more technically referred to as 'cracking'—of the asphaltene structure is the first step in making the oil

more accessible for further treatment [35–37]. Catalytic cracking also distributes the asphaltene aromatic structure into lighter fractions, which increases the value of the oil. Involving nanotechnology in the field of heavy oil recovery is a way of exploring efficient ways to implement the same process but with improved results [39].

Although so far, many nanomaterials have been developed to improve HO recovery, there is still work to be done to improve the quality of products obtained during the cracking of heavy oil fractions. Well-designed nanoparticles can achieve this goal, which should present a high affinity for heavy oil fractions (asphaltenes), that subsequently can be decomposed into lower molecular weight hydrocarbons and high calorific gases (such as hydrogen and others) by the interactions between steam and the catalytic active sites of nanoparticles [19,27,40–49].

In this context, this study looks for an alternative to implement energy transition strategies. It is well known that renewable energy sources should incorporate traditional energy sources to be more sustainable [7]. Hence, the application of tailor-made nanofluids for the revaluation and production of HO and EHO, in parallel, will entail obtaining $H_2$ as a transitory and complementary source of energy that will help the implementation of this fuel on a large scale until it achieves the development of 100% "eco" technologies that allow a sufficient supply of green $H_2$. However, the particular properties of hydrogen, such as the small size of the molecule, provide it with great transport capacity in a porous medium, even with almost impermeable properties [50]. Thus, it is imperative to analyze the thermodynamic characteristics of the $H_2$ produced in the reservoir during the implementation of nanotechnology-assisted steam injection.

To this end, this study considers both experimental and theoretical components. The experimental section includes the static and dynamic evaluation of nanoparticles during steam injection, considering two representative Colombian oil fields. For static experiments, two nanoparticles were considered. NP1: commercial $Al_2O_3$ nanoparticles (Petroraza S.A.S, Medellín, Colombia) doped with 1.0% in mass fraction of Ni and Pd; and NP2: commercial $CeO_2$ nanoparticles (Nanostructured & Amorphous Materials, Houston, TX, USA) doped with 1.0% in mass fraction of Ni and Pd. The best one was selected to design the nanofluid and perform the dynamic tests. Some of the analyzed characteristics include crude oil recovery, crude oil upgrading, and perdurability of the oil quality and produced gases. Next, a thermodynamic analysis of the fugacity of hydrogen was performed to obtain a clearer landscape of its in situ behavior. Based on this analysis, it was possible to determine the tendency of hydrogen to be trapped in the reservoir and its dissipation into the porous media.

## 2. Results

### 2.1. Nanoparticle Selection through Adsorption Isotherms and Thermogravimetric Analysis

Adsorption isotherms constructed for the *n*-$C_7$ asphaltenes isolated from HO1 and HO2 over NP1 y NP2 are shown in Figure S1 of the supplementary information. According to the International Union of Pure and Applied Chemistry (IUPAC), the adsorption isotherms profiles correspond to a type Ib, which agreed well with results reported previously [48], where the adsorption of asphaltenes on solid surfaces in nanometric sizes is described. In general, the asphaltene adsorption of heavy oil 1 (HO1) was slightly higher in nanoparticle 1 (NP1) and very similar to that obtained by nanoparticle 2 (NP2). The same train was found for the heavy oil 2 (HO2) asphaltenes. These results indicate a high affinity for both asphaltenes for the metallic phases of Ni and Pd and slightly higher for species based on alumina than on ceria. Figure S2a,b shows the non-isothermal thermogravimetric analysis at high pressure for asphaltenes adsorbed and non-adsorbed over NP1 and NP2. Panel A shows the results of asphaltenes isolated from HO1. Asphaltenes present the main decomposition peak at 510 °C and finish their decomposition at 590 °C. In this same figure, when asphaltenes are adsorbed on nanoparticles, their gasification seems to occur at much lower temperatures. The results indicate that the asphaltene

decomposition temperature is reduced from 510 °C to 200–250 °C. However, the rate of mass change profiles has different peaks associated with the different molecular weights of adsorbed asphaltenes and nanoparticle chemical nature (alumina and ceria). The alumina nanoparticle finishes the decomposition around 30 °C earlier than the ceria nanoparticles. For asphaltenes isolated from HO2, it is noted that they decompose at 500 °C in the absence of nanoparticles and around 210 °C, and 220 °C when they are adsorbed over NP1 and NP2, respectively. Finally, Figure S3 shows the isothermal conversion for both asphaltenes. NP1 decomposes 100% of both adsorbed asphaltenes at a lower time than NP2. Using NP1, asphaltenes from HO1 and HO2 are completely decomposed at 88 and 95 min, respectively.

On the other hand, Figure S4 shows resin adsorption isotherms for both crude oils (Panels A and B). The nanoparticles exhibit a type Ib adsorption isotherm for resins adsorption. For HO1 and HO2, NP1 uptake was higher than NP2. The difference in resins adsorption between NP1 and NP2 is around 0.23 (resins from HO1) and 0.25 mg m$^{-2}$ (resins from HO2), respectively. Compared with asphaltene adsorption, nanoparticles adsorb a similar amount of resins, which indicates good selectivity for both heavy compounds. Figure S5 shows the conversion of resins evaluated at isothermal conditions. The profiles show that resins conversion achieves 100% when nanoparticles catalyze the reaction; otherwise, just 30% of resins can be converted at the evaluated conditions. The time to decompose 100% of adsorbed resins increases in the order NP2 < NP1 for both samples, following the same trend of asphaltenes. All these results highlight the NP1 capacity to absorb and decompose asphaltene and resins over NP2.

## 2.2. Crude Oil Recovery

Based on static results, the nanofluid was formulated with NP1. The crude oil recovery curves for the dispersed nanofluid injection in the steam stream in HO1 and HO2 are shown in Figure 1a,b. The absolute permeability was estimated at 4331 mD and 2103 mD for HO1 and HO2 systems. Additionally, the oil effective permeability was 3558 and 1887 for the same systems. For the HO1, after the injection of 11 PVWE, an oil recovery of 54% was obtained. Some mechanisms are associated with the steam effect on crude oil production, including heat transfer to the rock and reservoir fluids, thermal expansion, volatilization of lighter hydrocarbons, and the disintegration of the viscoelastic network of crude oil. During the dispersed injection of the nanofluid into the steam, a 10% increase in recovered oil was obtained. This is mainly due to the higher contact area acquired by the tiny liquid droplets dispersed in the steam stream. Finally, after the last steam injection, an oil recovery of 81% was achieved.

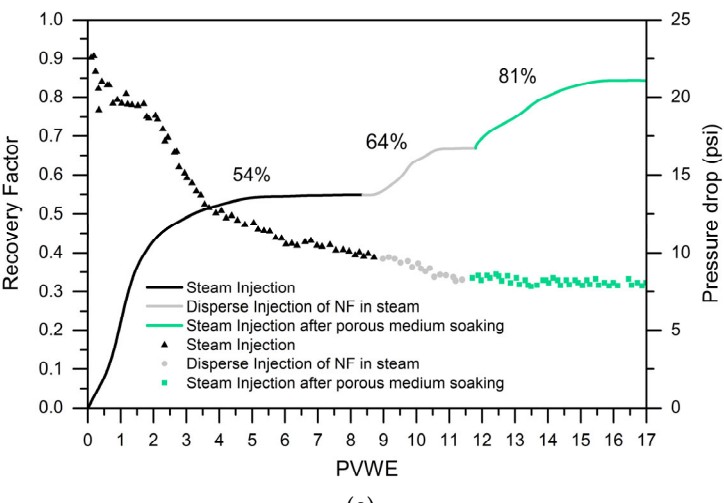

(a)

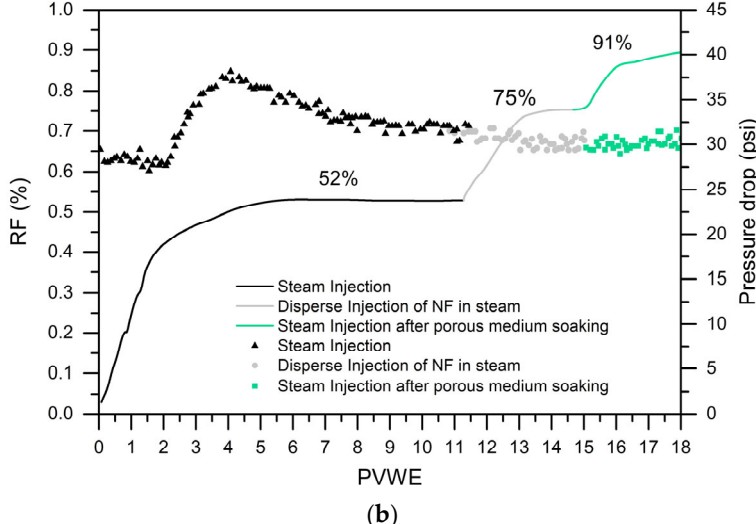

**Figure 1.** Oil recovery curve for steam injection assisted by Al-NF dispersed in the steam stream during the stages: (1) continuous steam injection, (2) dispersed injection of nanofluid in steam, and (3) steam injection after porous medium soaking in (**a**) HO1, and (**b**) HO2. Steam injection temperature: 270 °C for both cases. Pore pressure = 150 psi for both cases. Overburden pressure 1528 and 1992 psi for HO1 and HO2, respectively.

From Panel B in Figure 1, the crude oil recovery curve for HO2 is observed. During the first stage (steam injection without nanofluid), an oil recovery of 56% was obtained. Then, during the injection of nanofluid dispersed in the steam stream, the crude oil produced increased by 23%. Finally, 91% of the original oil in place (OOIP) was recovered at the end of the third stage. The missing 9% is considered residual oil saturation, which could not be displaced by the number of pore volumes of water equivalent injected (PVWE) and with similar characteristics to the untreated crude oil.

Instantaneous oil production was observed once the nanofluid was injected into the steam stream. This result potentiates the conventional steam injection technology and helps obtain better yields than in the scenarios assisted by injection with a liquid batch of nanofluid [19,51]. Some properties that can influence this process are the interactions between steam and the Al-, Ni-, and Pd- active sites of the nanoparticles [52]. Additionally, the small size of the injected nanofluid droplets impacts a large penetration radius, producing a higher recovery factor. This behavior is obtained for both crude oils during the second stage. It is expected that once the nanoparticles come into contact with the crude oil matrix, the phenomena of adsorption and catalysis of heavy fractions occur quickly, improving the mobility conditions of the crude oil. This result is consistent with the results obtained in previous evaluations, where an immediate recovery of an extra-heavy crude oil was obtained during the injection of a $CeO_2$-based nanofluid dispersed in the steam stream [39].

*2.3. Effluent Analysis*

Panels A and B of Figure 2 show the API gravity values for untreated crude oil, crude oil after steam injection, crude oil recovered by nanofluid injection dispersed in the steam stream, and crude oil after a soaking time of 12 h for HO1 and HO2, respectively. For HO1, the API gravity increased after steam injection without the nanoparticles from 12.4° to 12.6°. The results showed increases in the API up to 18° and 18.5° before and after the soaking treatment (Panel A). Similarly, for HO2, API gravity remained constant after steam injection without NF (Panel B). Then, during the nanofluid injection, the API gravity increased to 29° and after the soaking stage, it increased to 29.2°. To understand the difference in API gravity changes, it is essential to analyze the compositional changes in

the content of the SARA fractions. These results are shown in Panels C and D for HO1 and HO2, respectively. No appreciable or significant change in asphaltene content is observed during the first steam injection. The distribution of the rest of the SAR components is similar in both crude oils. This result agrees well with the unchanged API values described above. Likewise, several works in the literature report that steam does not modify the chemical composition of crude oil [15,30].

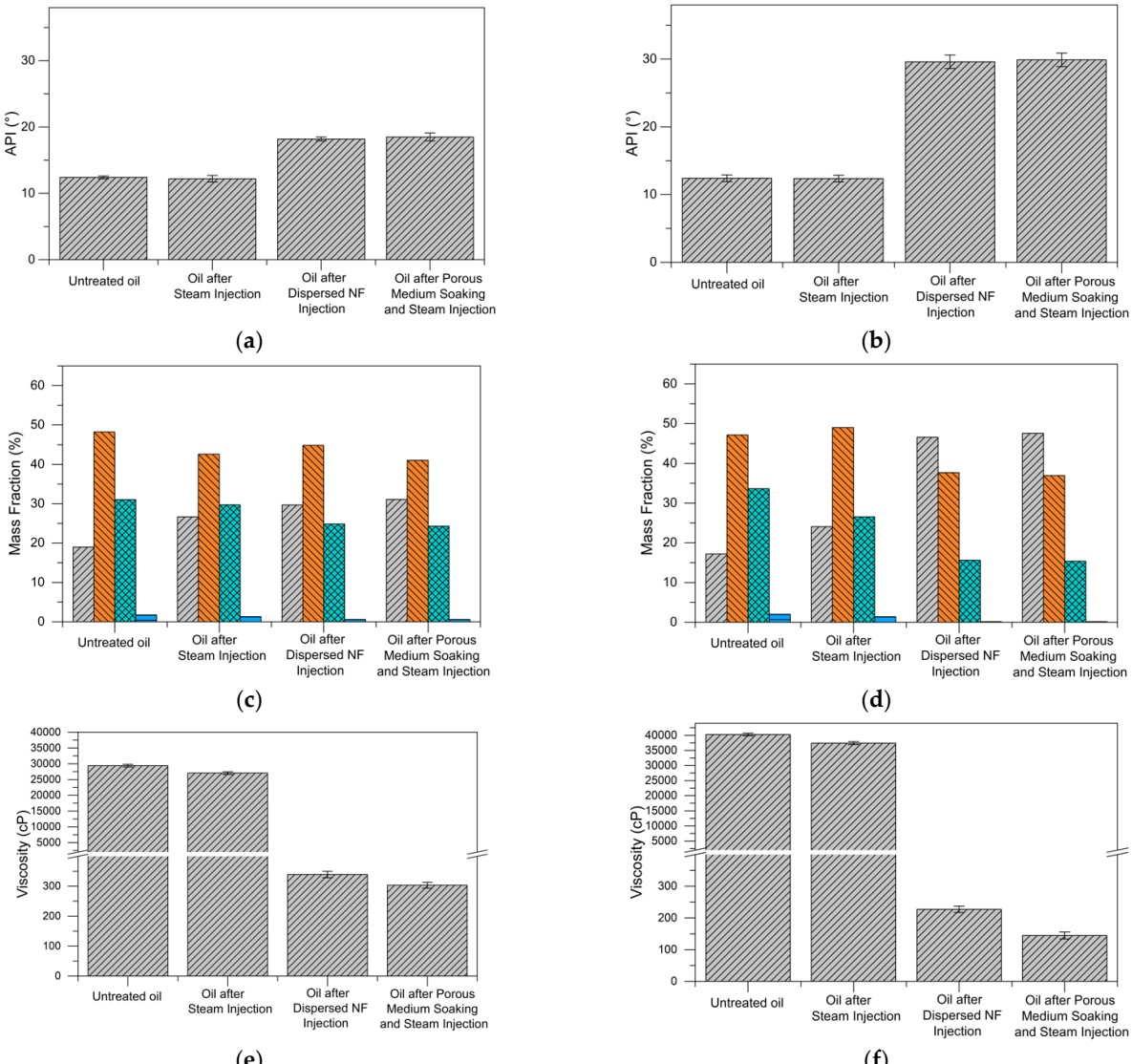

**Figure 2.** (**a**,**b**) API gravity and (**c**,**d**) SARA content distribution in wt.% (gray is saturates, orange is aromatics, green is resins and blue is aromatics) and (**e**,**f**) crude oil viscosity for untreated extra-heavy oil and crude oil recovered after the steam injection, during the injection of Al-NF dispersed in a steam stream, and after soaking for (**a**,**c**,**e**) HO1 and (**b**,**d**,**f**) HO2.

The presence of the NF generates a decrease from 1.76% to 0.7% and 0.2% of the asphaltene content in the HO1 for the respective stage after steam injection with nanofluid dispersed in its stream and subsequent steam injection after the 12 h soaking of the nanofluid with the porous medium, demonstrating the high catalytic activity of the catalyst. In the case of the HO2, the asphaltene content was reduced from 2.01% to 0.3% and 0.1% in a mass fraction in the same stages. Both crude oils had a very low final asphaltene content. However, the saturates and resins distribution change to a greater extent between

both samples. In the HO1, saturates increased from 19% (stage 1) to 26% (stage 3) in mass fraction, whereas in HO2, they increased from 17% (stage 1) to 46% (stage 3) in mass fraction. Finally, resin content was reduced by 22% (untreated crude oil) and 50% (crude oil recovered after soaking of NF and steam injection) for HO1 and HO2, respectively. According to these results, the injected NP1 attack both asphaltenes and high molecular weight resins, increasing the content of lighter hydrocarbons such as saturates. In this way, better quality is obtained in the HO2. Many factors can explain the good performance of NP1. First, the combined selectivities and reactivities of Ni and Pd toward the asphaltene and resin molecules result in a decrease in the content of both fractions. Additionally, because of strong metal support interactions (SMSI) alumina nanoparticles avoid metal sintering after the doping process, which leads to an increase in the number of active sites available for gasification reactions. Finally, the species –O and –OH resulting from the dissociative adsorption of steam by the alumina lower valence state, can be transferred to nickel and palladium and react with surface carbonaceous species [46]. Besides, through the movement of oxygen vacancies formed by the change in the oxidation state of the alumina species and the destabilization of the same, the reagents are transferred to the active sites of the transition element oxides.

Finally, Panels E and F depict the viscosity values at a shear rate of 10 s$^{-1}$ and 25 °C of the HO1 and HO2, respectively. During the first stage, oil viscosity was slightly reduced for both samples due to the reduction in cohesive intermolecular forces between asphaltenes and resins.

For the nanotechnology-assisted scenarios (second and third stage), a significant reduction in oil viscosity was noted, which was higher for the effluent recovered after 12 h of soaking. The oil viscosity for HO1 and HO2 recovered in the third stage was 300 and 104 cP, respectively. The main mechanism that explains the reduction in oil viscosity is the cracking/redistribution of asphaltene–resin systems and their subsequent stabilization through free radical hydrogenation to prevent the formation of heavier compounds.

Interestingly, crude oil upgrading was more noticeable in HO2 than in HO1. The catalytic activity of the material promoted higher API values and lower viscosities in the HO2 sample, probably due to interactions with its heaviest fractions. The chemical nature of the asphaltenes and resins of each crude oil greatly influences the response in gasification reactions for oil upgrading. In this sense, it is to be expected that the HO2 fractions are energetically easier to transform into lighter compounds.

On the other hand, Figure 3 shows the results of API and dynamic viscosity of the HO1 and HO1 recovered after the 12 h soaking of the nanofluid in the porous medium during days 1, 8, 15, and 30 after its recovery. In both systems, it is observed that API gravity and oil viscosity remain constant during the first 30 days evaluated, indicating the potential of the nanofluid to generate a permanent crude oil upgrading.

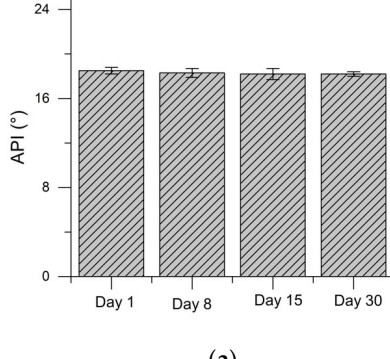
(**a**)

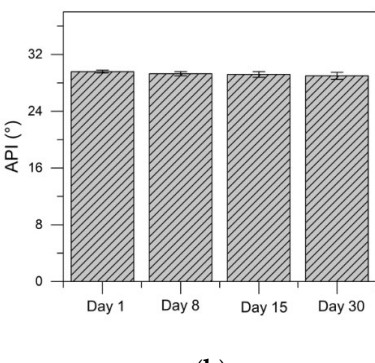
(**b**)

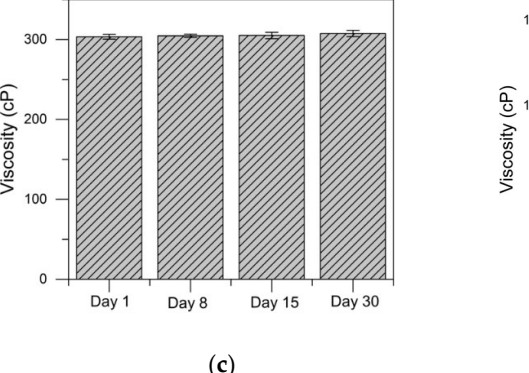

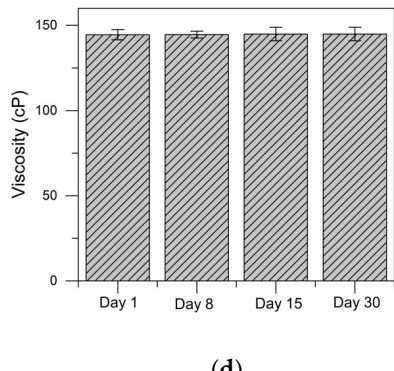

(**c**)　　　　　　　　　　　　　　　　　(**d**)

**Figure 3.** Perdurability of crude oil for (**a**,**c**) for HO1 and (**b**,**d**) for HO2, for the effluents recovered after the 12 h soaking of the nanofluid in the porous medium.

*2.4. Gaseous Products*

The gas produced during the dynamic test was collected in hermetically sealed aluminum containers for subsequent analysis in a mass spectrometer. The gases were analyzed during the first and third stages (i.e., steam injection and after NF soaking). The gaseous products released were $H_2$, $CO$, $CH_4$, $CO_2$, $C_2H_4$, and a small amount of $H_2S$. The volume fraction indicates the content of the components in the gaseous product, and the results are shown in Figure 4. Panel A in Figure 4 shows the results obtained for HO1. As the main result, it can be observed that the content of light hydrocarbons ($C_2H_4$ and $CH_4$) increases considerably after the injection of the nanofluid in the porous medium, accompanied by the reduction in $CO_2$ (<7% vol), $CO$ (<12% vol) and $H_2S$ (<1% vol). Additionally, the nanofluid generates $H_2$ during the catalytic cracking of the crude oil fractions, obtaining a gas with approximately 5% vol and 22% vol hydrogen during the first and third stages, respectively. The results obtained for HO2 are shown in Panel B of Figure 4. Similar components to HO1 are observed. The content of light hydrocarbons ($C_2H_4$ and $CH_4$) and hydrogen increased in the stage after the injection of the nanofluid. Hydrogen released was around 23% vol and 7% vol for the third and first stages, respectively. Oppositely, the gases $CO_2$, $H_2S$, and $CO$ decrease during the steam injection after the nanofluid soaking.

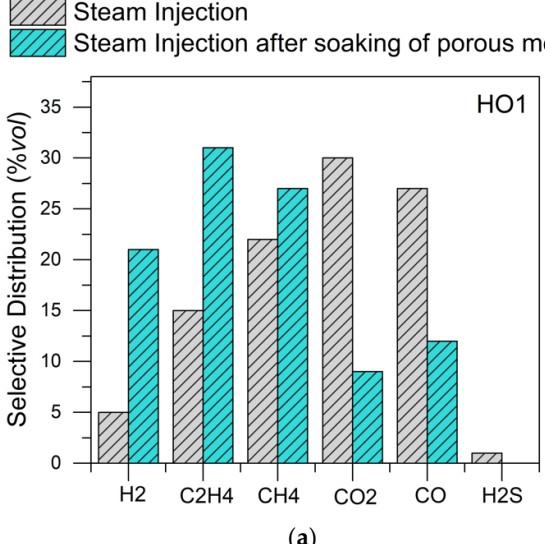

(**a**)

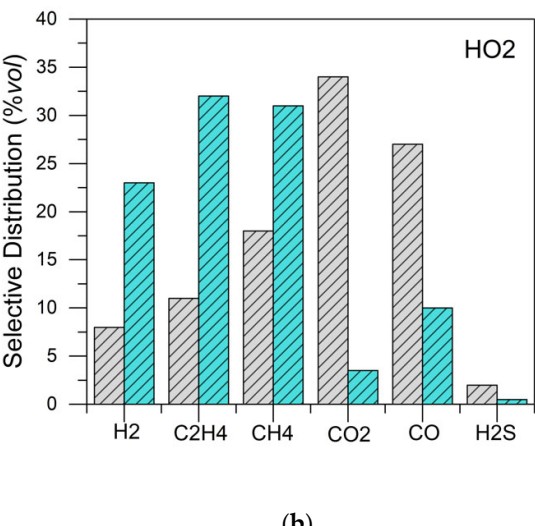

(**b**)

**Figure 4.** Selectivity distribution of light gases produced from steam gasification of (**a**) HO1 and (**b**) HO2 at reservoir conditions in the presence and absence of nanofluid.

The benefit of nanoparticles in heavy oil gasification is elucidated when hydrogen production is increased, and $CO_2$ release is reduced after the catalytic process. Hydrogen is a typical product of free radical reactions of the heavier molecules in crude oil. Therefore, the nanoparticles are expected to promote these reactions under the conditions evaluated, increasing the amount produced. A previous study demonstrated that hydrogen could be produced from the catalytic steam gasification of asphaltenes and resins [34]. First, the reaction between $H_2O$ and C atoms in both fractions releases hydrogen as a direct byproduct [52]. Other reactions, including water–gas shift and steam reforming, were also evidenced by the authors [52]. Nanoparticles could facilitate the production of $H_2$ from $H_2O$-$CH_4$ and $H_2O$-CO through steam reforming and water–gas shift reactions, respectively [53,54].

Moreover, the presence of Ni and Pd phases and their interactions with $Al_2O_3$ support benefit the production of $H_2$. For example, $H_2$ can be produced by $Ni/Al_2O_3$ phases through the complete combustion of $CH_4$, $H_2O$, and CO reforming. On the other hand, the $Pd/Al_2O_3$ can simultaneously produce different species such as $H_2$ and CO. Both systems follow different reaction pathways because of the further transfer of electrons between the active phase to the support [55,56].

Comparing crude oil upgrading results, the production of hydrogen and light hydrocarbons ($C_2H_4$ and $CH_4$) is correlated with heavy oil upgrading results. The higher the API gravity of the recovered crude oil during the displacement test, the higher the selective distribution of $H_2$, $C_2H_4$, and $CH_4$ during the conversion process. The opposite was found for the viscosity. It means that HO2 produced a higher amount of $H_2$ and light hydrocarbons than HO1 because the nanoparticles presented the best performance in the in situ upgrading of HO2. Both trends highlight the relationship between oil upgrading and the release of a gas mixture with high calorific power.

*2.5. Thermodynamic Analysis of Produced Hydrogen*

2.5.1. Hydrogen Fugacity on HO1 during Steam Injection Assisted by Nanoparticles

The analysis of hydrogen fugacity was performed using the molar composition of the released gas mixture presented in Table 1.

**Table 1.** Molar composition of the gas mixture released during steam injection test of HO1 during the first steam injection (without nanoparticles) and the steam injection after nanofluid soaking with porous medium (with nanoparticles).

| Component | Without Nanoparticles | With Nanoparticles |
|:---:|:---:|:---:|
| $H_2$ | 0.05 | 0.22 |
| $C_2H_4$ | 0.15 | 0.33 |
| $CH_4$ | 0.22 | 0.27 |
| $CO_2$ | 0.30 | 0.07 |
| $CO$ | 0.27 | 0.11 |
| $H_2S$ | 0.01 | 0.00 |

With the thermodynamic properties shown in the Methods section and molar composition (Table 1) of the gas mixture, the fugacity of each component was calculated (Equations (1)–(15)) for both scenarios with and without the assistance of nanoparticles. The pressure range used for HO1 was 600 psi–1600 psi, obtained through reservoir pressure and overburden gradient, respectively. Figure 5 depicts the results at steam injection temperature (270 °C). Hydrogen fugacity was found at 36.12 psi and 158 psi without and with nanoparticles, respectively, which means that hydrogen acquired a higher chemical potential due to the presence of nanoparticles. The hydrogen fugacity close to 1600 psi was 85.0 psi and 370.5 psi for the same scenarios. The hydrogen produced in the mentioned pressure range presents higher fugacity close to the limits of the reservoir; hence, it will move forwards to areas of lower fugacity (far from the reservoir limit with the overburden pressure). The smallest difference in fugacity in the scenario without nanoparticles suggests that at reservoir pressure, hydrogen has a lower tendency to migrate by diffusion to the low-pressure point. Therefore, hydrogen can be lost in the stratigraphic columns above the reservoir. The results agree with those reported by Chen et al. [57]. The authors show the separation of $CH_4$ and $H_2$ with fugacity measurements, where high pressures allow the purification of the components using a membrane-like technology. In this way, they show the fugacity of each component as a tool to evaluate the separation and purification of components.

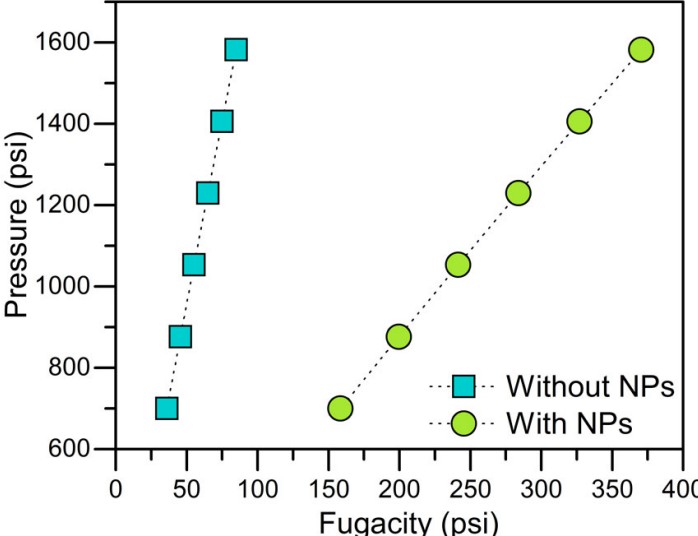

**Figure 5.** Fugacity of $H_2$ generated during the steam injection in the presence and absence of nanoparticles on HO1 between reservoir pressure (700 psi) and overburden pressure (1600 psi) at the steam injection temperature (270 °C).

### 2.5.2. Hydrogen Fugacity on HO1 at Different Temperatures Assisted by Nanoparticles

The temperature variation was selected according to the thermal gradient observed on the porous media of the displacement tests. Table 2 summarized the temperature of the coordinates. The pressure range was 700–1600 psi, as in the previous section.

**Table 2.** Temperature of coordinates of HO1 during the first steam injection (without nanoparticles) and the steam injection after nanofluid soaking with porous medium (with nanoparticles).

| Component | Temperature (°C) | |
| :---: | :---: | :---: |
| | **Without Nanoparticles** | **With Nanoparticles** |
| 1 | 270 | 270 |
| 2 | 212 | 255 |
| 3 | 168 | 234 |
| 4 | 59 | 150 |

Figure 6 shows the fugacity contour as a function of pressure and temperature for hydrogen produced during the steam injection non-assisted and assisted by nanoparticles. With the increase in temperature, fugacity decreased in both scenarios. When nanoparticles were used, fugacity increased from 158 psi to 161 psi between 1 and 4 coordinates at reservoir pressure. In the absence of nanoparticles, fugacity also increased by 3 psi in the same conditions. As pressure increased (close to the overburden pressure), the nanoparticles-assisted step increased fugacity at a higher degree when the temperature lowered from coordinate 1 to 4. In this way, when comparing both scenarios, nanoparticles increase the gap between the fugacities that the hydrogen acquires in the external points evaluated. This leads to reduced displacement of the gas out of the reservoir. Similar behavior of hydrogen within a gas mixture was shown by Redlich et al. [58].

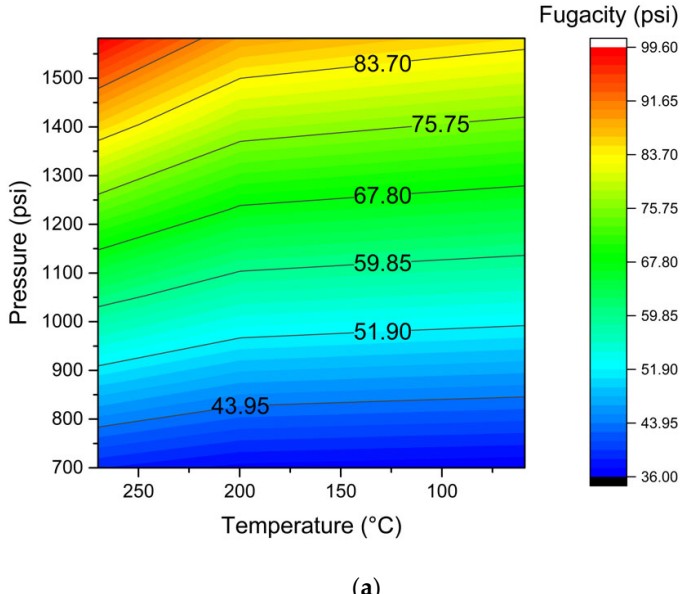

(**a**)

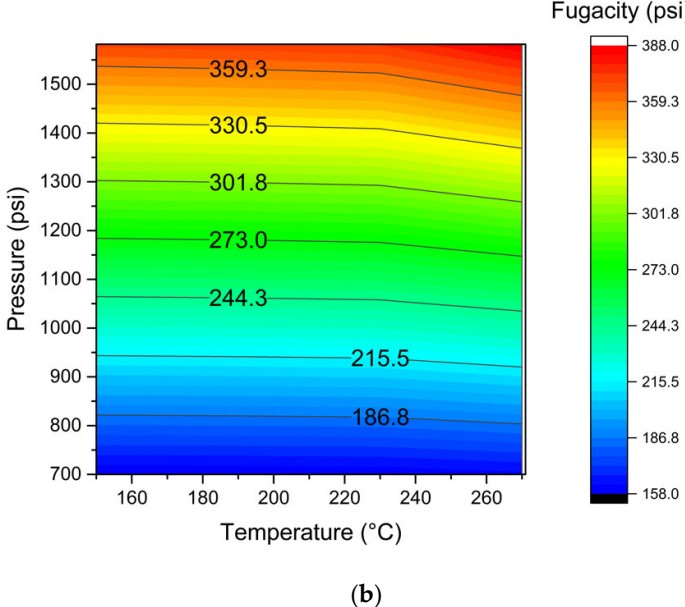

**(b)**

**Figure 6.** Fugacity contour of $H_2$ generated during the steam injection in the (**a**) absence and (**b**) presence of nanoparticles on HO1 between reservoir pressure (700 psi) and overburden pressure (1600 psi) as a function of temperature decline.

It is worth mentioning that at low pressure, fugacity acquires lower values (reservoir pressure) regardless of the temperature and scenario evaluated (with or without nanoparticles). This result could be explained because the fugacity is a correction to the relative pressure of the component in the mixture, and at higher pressures, there are more significant deviations due to the intermolecular interactions between the components of the gas mixture [59,60]. In this way, the effect of nanoparticles on hydrogen fugacity holds more importance at higher system pressures. All these results highlight the use of nanoparticles to assist a steam injection process where it is possible to produce hydrogen during heavy crude oil production.

### 2.5.3. Hydrogen Fugacity on HO2 at Different Temperatures in the Presence and Absence of Nanoparticles and Comparison with HO1

This section describes the estimated results of fugacity for HO2, considering the reservoir pressure (400 psi) and the overburden pressure (1992 psi) at the same temperature range evaluated in HO1. Figure 7 shows the contour areas for $H_2$ generated during the steam injection. The $H_2$ fugacity at reservoir pressure was 32.5 psi and 93.3 psi when $H_2$ was produced in the absence and presence of nanoparticles at 270 °C. Regarding overburden pressure, the fugacity of $H_2$ was around 173 and 210 psi for the same scenarios, respectively. Based on these results, it is expected that the hydrogen produced in zones closer to the overburden pressure will migrate more easily toward zones of lower pressure (i.e., around 400 psi). The difference in fugacity between the unassisted and nanoparticle-assisted scenarios is 141 and 400 psi. This means that the $H_2$ produced in the stage without nanoparticles tends to migrate more easily through the porous medium out of the reservoir concerning the hydrogen obtained when there are nanoparticles. The results agree with those obtained for HO1. The difference between both systems lies mainly in the fact that the pressure difference between the reservoir and the overburden pressure is greater in HO2. The difference in fugacity found is greater when the pressure differential is greater.

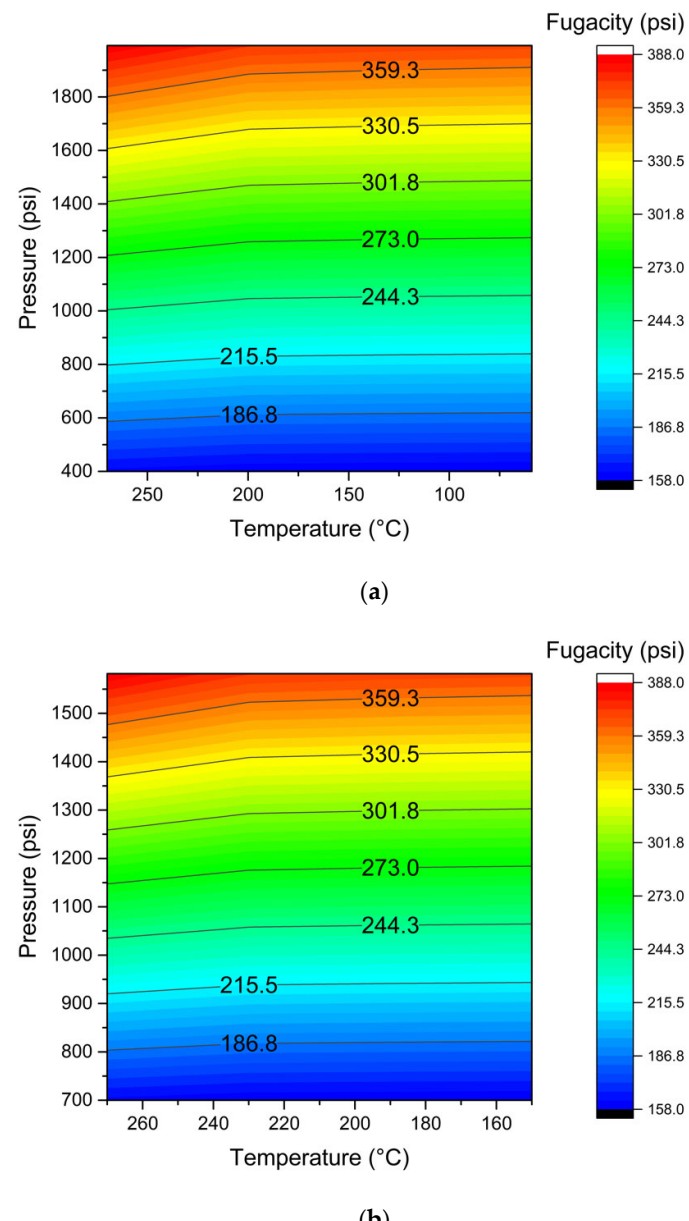

(**a**)

(**b**)

**Figure 7.** Fugacity contour of H$_2$ generated during the steam injection in (**a**) absence and (**b**) presence of nanoparticles on HO2 between reservoir pressure (400 psi) and overburden pressure (1990 psi) as a function of temperature decline.

## 3. Materials and Methods

### 3.1. Materials

Asphaltenes and resins were extracted based on ASTM D2892 and ASTM D5236 standards [61,62] from two different HO. *n*-Heptane (99% Sigma-Aldrich, St. Louis, MO, USA) was used as the precipitating agent. Static tests were executed with two different nanoparticles. NP1: commercial Al$_2$O$_3$ nanoparticles (Petroraza S.A.S, Medellín, Colombia) doped with 1.0% in mass fraction of Ni and Pd; and NP2: commercial CeO$_2$ nanoparticles (Nanostructured & Amorphous Materials, Houston, TX, USA) doped with 1.0% in mass fraction of Ni and Pd. Nickel and palladium were doped through the incipient wetness technique [63]. The doped amount of Ni and Pd was ensured by Energy Dispersive

X-ray spectroscopy (EDX) using a Field Electron and Ion (FEI) microscope model Quanta 400 (SEM) (Eindhoven, The Netherlands) coupled with the EDX source.

Both materials have been extensively characterized and reported by Cardona et al. [64] and Medina et al. [65]. The most important properties are shown in Table 3.

**Table 3.** Basic characteristic of nanoparticles.

| Properties | NP1 | NP2 |
|---|---|---|
| Hydrodynamic diameter (nm) | 76.0 | 20.2 |
| BET surface area ($m^2 \cdot g^{-1}$) | 223.4 | 65.4 |
| Ni crystal size (nm) | 2.2 | 6.4 |
| Pd crystal size (nm) | 4.1 | 3.9 |
| Ni dispersion (%) | 5.4 | 12.7 |
| Pd dispersion (%) | 9.9 | 38.6 |
| Point of zero charge | 7.8 | 7.5 |

Displacement tests were performed using the reservoir fluids of two different Colombian fields, consisting of two heavy crude oils and two synthetic brines. The basic properties of the crude oils are shown in Table 4. Brines were composed of 22,000 (brine 1) and 18,000 ppm (brine 2) NaCl eq (Sigma-Aldrich, St. Louis, MO, USA). Deionized water with 3 $\mu S \cdot cm^{-1}$ conductivity was used to generate the steam.

**Table 4.** Basic characteristics of heavy crude oil for the steam injection test.

| Properties | HO1 | HO2 |
|---|---|---|
| API° | 12.4 | 12.1 |
| Viscosity 25 °C | 4000 | 3500 |
| Saturates (%) | 18.98 | 17.18 |
| Aromatic (%) | 48.24 | 47.16 |
| Resins (%) | 31.04 | 33.65 |
| Asphaltenes (%) | 1.76 | 2.01 |

Sand samples were provided by Ecopetrol S.A. to construct the porous media. The constructed porous media for the two Colombian fields were cleaned with a mixture of methanol (99.8%), toluene (99.8%), and HCl (37%), all provided by Merk KGaA (Darmstadt, Germany) following the tests reported in previous studies [24,25].

The nanoparticle with the best performance in the static tests is selected to formulate the nanofluid. The carrier fluid consists of an oil-based fluid provided by Petroraza S.A.S (Medellín, Colombia). Fourier Transform–Infrared Spectra corroborated the oil-based chemical nature [66–68], and this is shown in Figure S4 of the supplementary material. The nanofluid (Al-NF) consists of 500 $mg \cdot L^{-1}$ nanoparticles dispersed in the commercial carrier. Some properties of the carrier and the nanofluid are shown in Table 5.

**Table 5.** Basic characteristics of carrier and nanofluid.

| Properties | Carrier | Nanofluid |
|---|---|---|
| Density ($g \cdot mL^{-1}$) | 0.96 | 0.96 |
| Viscosity (cP) | 2.13 | 3.05 |
| Surface tension ($mN \cdot m^{-1}$) | 24.23 | 23.01 |
| Conductivity ($mS \cdot cm^{-1}$) | 4.9 | 5.7 |
| Thermal conductivity ($W \cdot mK^{-1}$) | 0.1502 | 0.1562 |
| Thermal resistivity (°C $cm \cdot W^{-1}$) | 660 | 640 |

### 3.2. Static Tests for Nanoparticle Selection through Adsorption Isotherms and Thermogravimetric Analysis

The asphaltene/resin adsorption capacity of the nanoparticles was measured through the construction of adsorption isotherms preparing oil model solutions consisting of different concentrations of asphaltenes/resins (100 mg·L$^{-1}$–2000 mg·L$^{-1}$) diluted in toluene. The instrument and protocol employed for the adsorption isotherms construction were described in previous studies [39,65,69].

A high-pressure thermogravimetric analyzer evaluated the subsequent catalytic decomposition of asphaltenes/resins adsorbed on nanoparticles (HP-TGA 750, TA instruments Inc., Hüllhorst, Germany). The tests were carried out at 700 and 400 psi for the asphaltenes isolated from HO1 and HO2 based on the pressure of the respective oil field.

Initially, the surface of the samples was cleaned by subjecting them to a vacuum at 0.036 psi for 10 min. After that, the equipment reached pressure and flow conditions before warming up. The experiments were carried out at isothermal steam injection conditions (250 °C) for 300 min. The experiments were executed for an asphaltene/resin adsorbed amount of 0.2 mg·m$^{-2}$ ± 0.02 mg·m$^{-2}$ [70]. The steam atmosphere was simulated by introducing 100 mL·min$^{-1}$ of $N_2$ and 6.30 mL·min$^{-1}$ of $H_2O_{(g)}$ using a gas saturator controlled by a thermostatic bath [71].

### 3.3. Oil recovery and Upgrading Evaluation

For dynamic tests, two different porous media were used. Table 6 summarizes the absolute and oil-effective permeability for both systems.

**Table 6.** Basic characteristics of porous media.

| System | Porous Medium 1 | Porous Medium 2 |
|---|---|---|
| Mineralogy | Silica (99%) | Silica (99%) |
| Porosity (%) | 22.0 | 21.0 |
| Absolute permeability | 4331 | 2103 |
| Oil effective permeability | 3558 | 1887 |

Displacement tests were executed in three stages to recreate steam injection at field conditions. Steam was injected at 80% quality at 250 °C. The steam quality was ensured through numerical simulation using the protocol described in our previous works [39,64]. The first stage includes the steam injection between 3 mL·min$^{-1}$ and 5 mL·min$^{-1}$. During this stage, the steam was injected until no more crude oil was produced. Then, during the second stage, the incremental crude oil produced by nanofluid injection dispersed into the steam stream was estimated. The nanofluid was injected between 0.5 mL·min$^{-1}$ and 1 mL·min$^{-1}$. The third stage started when no increment in oil production was observed. Here, the porous media were left to stand for 12 h, and then the steam was injected again until there was no oil production. The pressure profile was monitored during the complete process to ensure the nanofluid transport in the steam stream.

The overburden pressure was fixed at 1582 psi and 1992 psi for HO1 and HO2, respectively. The pore pressure was 150 psi in both cases. The instrumental details of the configuration system are reported in previous works [21,39,64].

Finally, the gas outlet line was coupled with a mass spectrometer (Shimadzu MS, Tokyo, Japan). The scan rate of the linear ion trap mass analyzer was 0.03 *m/z* from 0 *m/z* to 200 *m/z*. The MS instrument was equipped with a hot capillary column heated at 150 °C to prevent gas condensation. The components targeted for analysis were obtained by using a 100-eV electron impact mode to achieve sufficiently strong signals for information on the HO transformation.

### 3.4. Effluents Characterization

The crude oil recovered during the three stages of the displacement test was characterized by different techniques, including SARA distribution following the ASTM D6560 standard [51,72]. Additionally, API gravity and dynamic viscosity were measured using an Anton Paar Stabinger SVM 3000 (Madrid, Spain), following the protocols described elsewhere [21,64]. The perdurability of the treatment in the crude oil quality was evaluated in terms of API gravity and dynamic viscosity for four consecutive weeks.

### 3.5. Description of the Considered Scenario for In Situ Hydrogen Generated

With the aim of analyzing the behavior of hydrogen in the reservoir during in situ crude oil upgrading, a thermodynamic analysis based on its fugacity was performed. Figure 8 shows the schematic representation of the proposed system. The system considers a scenario where both the producing well and the injection well are closed, and gases have been generated during in situ crude oil upgrading. The $\Delta P$ on the *x*-axis is neglected ($P_{fw,injection} = P_r = P_{wf,production}$), whereas the $\Delta P$ on the *y*-axis was calculated based on the overburden gradient and the reservoir pressure of each oil field. The overburden gradient is obtained considering the density of the rock matrix for the two reservoirs at 2.55 g·cm$^{-3}$ (sand rock) and the depth of the reservoirs (HO1 = 1600 ft, HO2 = 2000 ft). The overburden pressures for HO1 and HO2 were 1582 and 1992 psi, respectively. Additionally, the system considers a caprock below the reservoir, so hydrogen diffusion could only occur upwards.

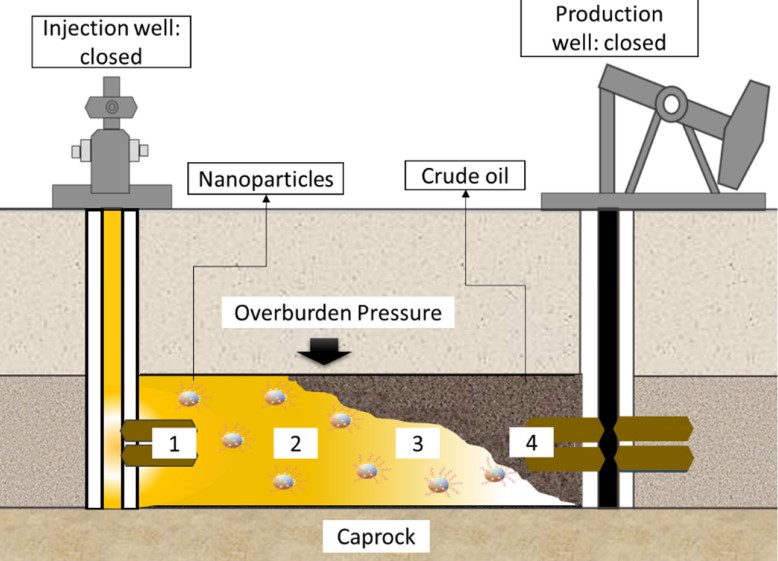

**Figure 8.** Schematic representation of considered scenario for hydrogen fugacity analysis. The system considers the following assumptions: injection and production well are closed (the $\Delta P$ on the *x*-axis is neglected); the overburden pressures for HO1 and HO2 were 1582 and 1992 psi, respectively; the density of the rock matrix for the two reservoirs was 2.55 g·cm$^{-3}$ (sand rock); the depths of the reservoirs were HO1 = 1600 ft, HO2 = 2000 ft; white boxes assume different temperatures based on displacement tests;and presence of caprock below the reservoir.

This section contemplates different analyses as follows:

1.  Hydrogen fugacity analysis at steam injection temperature (first box of Figure 8) for the gaseous mixture produced without nanoparticles on the HO1 sample.
2.  Hydrogen fugacity analysis at steam injection temperature (first box of Figure 8) for the gaseous mixture produced from the nanotechnology-assisted steam injection on the HO1 sample.

3. Hydrogen fugacity analysis as a function of temperature (second, third, and fourth boxes of Figure 8). The temperature was fixed based on the thermal profiles obtained in the experimental setup using four thermocouples at the beginning, inside, and exit of the porous medium. The analysis was performed for steam injection assisted by nanoparticles on HO1.

4. Hydrogen fugacity analysis for HO2 considering the variation in temperature and comparison with HO1.

### 3.6. Thermodynamic Analysis of In Situ Hydrogen Generated

Fugacity is a thermodynamic property that measures the chemical potential of species and can be used in phase equilibrium calculations. The use of fugacity allows the identification of the phase in which a component is likely to remain [60]. Its analysis can be conducted through cubic [73,74] or virial [59,75] equations of state in a mixture of gases or through correlations obtained from experimental information [57,76]. For this research, the fugacity of the gas mixture released during the displacement test of HO1 and HO2 by steam injection assisted by nanotechnology was calculated. The gas mixture contains hydrogen ($H_2$), ethylene ($C_2H_4$), methane ($CH_4$), carbon dioxide ($CO_2$), carbon monoxide (CO), and hydrogen sulfide ($H_2S$).

The application of virial equations of state truncated in the second term with mixing rules was considered to represent the possible intermolecular interactions between pairs of components. The truncation to the second term occurs because the interaction between pairs of molecules in gases is more likely than between triples or higher-order interactions [60,77].

The mixing rules allow us to calculate the properties of pseudo components that represent the interaction between pairs of molecules. The mixing properties considered were the acentric factor $(\omega)$, critical temperature $(T_c)$, critical compressibility factor $(Z_c)$, critical molar volume $(v_c)$, and critical pressure $(P_c)$. The mixing rules are applied to the critical conditions because they are characteristic of each compound and allow the evaluation of the deviation from ideality from the concept of reduced property, which is explained by the theorem of corresponding states [78].

The equations used to calculate the fugacity of gases using the virial equations of state truncated in the second term are shown below. First, the mixing rules are shown, then the calculation of the virial coefficients of the pure substances, as well as the coefficients of the interactions.

Next, we provide the mixing rules used to calculate the mixing properties of gases for the application of the truncated virial equation of state in the second term (Equations (1)–(5)).

$$\omega_{ij} = \frac{\omega_i + \omega_j}{2} \tag{1}$$

$$T_{cij} = \left(T_{ci}T_{cj}\right)^{1/2} \tag{2}$$

$$Z_{cij} = \frac{Z_{ci} + Z_{cj}}{2} \tag{3}$$

$$v_{cij} = \left(\frac{\left(v_{ci}\right)^{1/3} + \left(v_{cj}\right)^{1/3}}{2}\right)^3 \tag{4}$$

$$P_{cij} = \frac{Z_{cij}RT_{cij}}{v_{cij}} \tag{5}$$

where the subscript *ij* indicates a property of the mixture and the subscripts *i* and *j* indicate the component *i* and component *j*, respectively.

Thus, Equation (6) was used to calculate the second virial coefficient for the component mixture.

$$B_{ij} = \frac{RT_{cij}\hat{B}_{ij}}{P_{cij}} \tag{6}$$

where $\hat{B}_{ij}$ is the reduced second virial coefficient. To calculate the reduced virial coefficient, it must be considered that this is only a function of temperature [60]. A good approximation to this property is the following equation (Equation (7)):

$$\hat{B}_{ij} = B_{ij}^0 + \omega_{ij}B_{ij}^1 \tag{7}$$

where $B_m^0$ y $B_m^1$ are only a function of temperature. These values can be calculated through the following equations (Equations (8) and (9)):

$$B_{ij}^0 = 0.083 - \frac{0.422}{T_{rij}^{1.6}} \tag{8}$$

$$B_{ij}^1 = 0.139 - \frac{0.172}{T_{rij}^{4.2}} \tag{9}$$

where $T_{rij}$ refers to the reduced temperature of the mixture and is calculated as the ratio of the system temperature $T$ to the critical temperature of the mixture $T_{cij}$ (Equation (10)).

$$T_{rij} = \frac{T}{T_{cij}} \tag{10}$$

For the second virial coefficient calculation, the reduced properties were considered. To carry out the calculations previously explained, it is necessary to consider the properties of each species in the gas mixture. For this purpose, the following table (Table 7) shows the properties of interest for each of the components considered.

**Table 7.** Thermodynamic properties of gas mixture components produced during the steam gasification of HO1 and HO2 with and without nanoparticles.

| Component | Critical Temperature (K) | Critical Pressure (bar) | Critical Molar Volume (cm³·mol⁻¹) | Critical Compressibility Factor | Acentric Factor |
|---|---|---|---|---|---|
| $H_2$ | 33.19 | 13.13 | 64.1 | 0.305 | −0.216 |
| $C_2H_4$ | 282.3 | 50.4 | 131 | 0.281 | 0.087 |
| $CH_4$ | 190.6 | 45.99 | 98.6 | 0.286 | 0.012 |
| $CO_2$ | 304.2 | 73.83 | 94 | 0.274 | 0.224 |
| $CO$ | 132.9 | 34.99 | 93.4 | 0.299 | 0.048 |
| $H_2S$ | 373.5 | 89.63 | 98.5 | 0.284 | 0.094 |

The second virial coefficient is a function of composition and temperature in a gas mixture. A composition dependence equation at moderate pressures is as follows (Equation (11)) [60]:

$$B = \sum_i \sum_j y_i y_j B_{ij} \tag{11}$$

To obtain a mathematical expression for calculating the fugacity, it is necessary to develop the expression from the virial equation of state (Equation (12)).

$$Z = 1 + \frac{BP}{RT} \qquad (12)$$

Equation (12) can be rewritten for a mixture of n moles as Equation (13):

$$Zn = n + \frac{BPn}{RT} \qquad (13)$$

In this way, considering an expression for the coefficient of fugacity of component $k$ $\left( \hat{\phi}_k \right)$ in the mixture (Equation (14)), an expression for the activity coefficient of component $k$ in the gas mixture is obtained from the virial equation of state (Equation (15)):

$$\ln \hat{\phi}_k = \int_0^P \left( \overline{Z_k} - 1 \right) \frac{dP}{P} \qquad (14)$$

$$\ln \hat{\phi}_k = \frac{P}{RT} \left[ B_{kk} + \frac{1}{2} \sum_i \sum_j y_i y_j \left( 2\delta_{ik} - \delta_{ij} \right) \right] \qquad (15)$$

where subscripts $i$ and $j$ represent all species and $\delta_{ij}$ are calculated as shown below (Equations (16) and (17)):

$$\delta_{ik} = 2B_{ik} - B_{ii} - B_{kk} \qquad (16)$$

$$\delta_{ij} = 2B_{ij} - B_{ii} - B_{jj} \qquad (17)$$

The calculations were carried out using the Matlab® software (Version R2021a, Mathworks Inc., Natick, Massachusetts, USA.).

The purpose of the calculations made here is to show how the chemical potential of hydrogen is affected by the presence of nanoparticles, modifying its behavior.

## 4. Conclusions

The present work evidence, for the first time, the positive thermal effect of nanoparticles in assisting a steam injection process in terms of upgraded crude oil and hydrogen co-production. The fugacity of $H_2$ was determined between the reservoir and overburden pressure and different temperatures, which were determined by the thermal profiles in the displacement test. The fugacity was calculated using the application of virial equations of state with mixing rules based on the possible intermolecular interactions between the components. Hydrogen acquired a higher chemical potential due to the nanoparticles' presence. However, the difference in $H_2$ fugacity between both points is much higher with nanoparticles, which means that hydrogen presents a lower tendency to migrate by diffusion to the high-pressure point. The difference between HO1 and HO2 lies mainly in the fact that the pressure difference between the reservoir and the overburden pressure is greater in HO2; therefore, the difference in fugacity is greater when the pressure differential is greater. By considering the fugacity of each species as a measure of the chemical potential, the development of this work allows us to elucidate the effect of nanoparticles on the fugacity of the hydrogen formed in the reservoir due to thermal treatments, providing a clearer landscape of in situ hydrogen behavior, as well as the possibility of having a gaseous mixture rich in $H_2$ on the surface with upgraded crude oil.

**Supplementary Materials:** The following supporting information can be downloaded at: https://www.mdpi.com/article/10.3390/catal12111349/s1, Figure S1: Experimental adsorption of *n*-$C_7$ asphaltene isolated from (a) HO1 and (b) HO2 over NP1 and NP2 (dotted lines represent the SLE fitting). Figure S2: Rate for mass change profiles for steam gasification of *n*-$C_7$ asphaltenes isolated form (a) HO1 and (b) HO2, with and without NP1 and NP2. Heating rate: 10 °C·min⁻¹, $N_2$ flow: 100 mL·min⁻¹, $H_2O_{(g)}$ flow: 6.7 mL·min⁻¹and sample mass 1 mg. Figure S3: Isothermal conversion for steam gasification of *n*-$C_7$ asphaltenes isolated form (a) HO1 and (b) HO2, with and without NP1



and NP2. Heating rate: 10 °C·min⁻¹, N₂ flow: 100 mL·min⁻¹, H₂O₍g₎ flow: 6.7 mL·min⁻¹and sample mass 1 mg. Figure S4: IR spectra for carrier used in the nanofluid for steam injection displacement.

**Author Contributions:** Conceptualization, O.E.M., F.B.C. and C.A.F. (Camilo A. Franco); methodology, O.E.M. and S.C.; formal analysis, O.E.M.; investigation, O.E.M.; data curation, O.E.M., and S.C.; writing—original draft preparation, O.E.M.; writing—review and editing, all authors; supervision, C.A.F. (Camilo A. Franco) and A.F.P.-C. All authors have read and agreed to the published version of the manuscript.

**Funding:** This research received no external funding.

**Data Availability Statement:** Not applicable.

**Acknowledgments:** The author thanks the Grupo de Investigación en Fenómenos de Superficie-Michael Polanyi.

**Conflicts of Interest:** The authors declare no conflicts of interest.

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
