# Peer review of "A Theoretical and Experimental Approach to the Analysis of Hydrogen Generation and Thermodynamic Behavior in an In Situ Heavy Oil Upgrading Process Using Oil-Based Nanofluids"

_catalysts, doi:10.3390/catal12111349_

Round 1

Reviewer 1 Report

In the present work, a theoretical and experimental approach to the analysis of hydrogen generation were built and its thermodynamic behavior in an in-situ upgrading process of heavy crude oil using nanotechnology was studied. Two nanoparticles were evaluated in asphaltene adsorption/decomposition under a steam atmosphere. Then, a nanofluid containing nanoparticles on a light hydrocarbon is formulated and injected in a dispersed form in the steam stream during steam injection recovery tests of two Colombian heavy crude oils. The nanoparticles increased the oil recovery by 27% and 39% for two oils regarding the steam injection. The oil recovery at the end of the displacement test was 85% and 91% for two oils. After consideration, I recommend its publication in Catalysts after major revision. Here are my detailed comments:

(1) In the Introduction section, it is too simple for the introducing of the nanoparticles which can catalyze the cracking of heavy oil components. In addition, the application status of nanoparticle fluid and the problems encountered are not mentioned.

(2) The structural characterization of inorganic nanoparticle catalyst is too simple. Usually, XRD, XPS and Raman spectroscopy are used to analyze the crystal structure of inorganic particles. In addition, how to doping the original Al2O3 and CeO2 with Ni and Pd needs to be explained. How can the doping be proved to be successful?

(3) The author only analyzed the gas products after the cracking of two kinds of crude oil, but did not study the liquid products in depth.

(4) The authors have not to study the catalytic mechanism of the catalyst, such as what is the catalytic center? What is the relationship between catalytic efficiency and catalyst structure? What crude oil components can be cracked by the catalyst? What is the cracking mechanism of these components?

(5) It can be seen that the resin content of both crude oils is much higher than that of asphaltene. Is the main object of catalytic cracking resins? What is the reaction equation for its cracking?

(6) Although fugacity calculation has been used to study the effect of hydrogen, the experimental data support is still not strong enough.

Author Response

Medellín, October 26 2022

Ms. Stefania Pop
Assistant Editor

MDPI

RE: Response to comments of the reviewer 1 regarding manuscript ID: catalysts-1971815

Dear Ms. Stefania Pop,

We would like to thank you for securing a prompt review of our manuscript entitled; “A Theoretical and Experimental Approach to the Analysis of Hydrogen Generation and Thermodynamic Behavior in an In-Situ Heavy Oil Upgrading Process using Oil-Based Nanofluids”.  We have addressed all the comments raised by the reviewers and have thoroughly revised the manuscript accordingly. We found the comments helpful and believed that our revised manuscript represents a significant improvement over our initial submission.

The detailed response (in blue) to the reviewers’ comments, suggestions, and questions (in black), and the revised manuscript are attached. As suggested, any track changes, highlights, or font colors in our revised manuscript have been removed, and we believe now that our manuscript is publishable in the Catalysts journal.  

Please do not hesitate to contact us if you have any further questions.

Sincerely yours,

The authors

Responses to the reviewer´s suggestions for the manuscript:  A Theoretical and Experimental Approach to the Analysis of Hydrogen Generation and Thermodynamic Behavior in an In-Situ Heavy Oil Upgrading Process using Oil-Based Nanofluids, Catalysts.

Comment: In the present work, a theoretical and experimental approach to the analysis of hydrogen generation were built and its thermodynamic behavior in an in-situ upgrading process of heavy crude oil using nanotechnology was studied. Two nanoparticles were evaluated in asphaltene adsorption/decomposition under a steam atmosphere. Then, a nanofluid containing nanoparticles on a light hydrocarbon is formulated and injected in a dispersed form in the steam stream during steam injection recovery tests of two Colombian heavy crude oils. The nanoparticles increased the oil recovery by 27% and 39% for two oils regarding the steam injection. The oil recovery at the end of the displacement test was 85% and 91% for two oils. After consideration, I recommend its publication in Catalysts after major revision.

Response: We thank the reviewer for his/her effort to ensure a prompt revision of the manuscript, as well as his/her interest in improving its quality. Based on the comments, we have improved the manuscript. Below is the point-by-point response:

Issue 1: In the Introduction section, it is too simple for the introducing of the nanoparticles which can catalyze the cracking of heavy oil components. In addition, the application status of nanoparticle fluid and the problems encountered are not mentioned.

Response: We thank the suggestion. Based on that, the following was added to the introduction section.

“Consequently, nanoparticles have been extensively explored in the field of heavy oil recovery assisting conventional thermal treatments like a steam injection. A clear understanding of how nanoparticles interact with crude oil is an area of extensive research. Authors have made big efforts on understanding parameters such as the rheology of heavy oil as well as compositional changes to obtain insights on how crude oil upgrading can be done [18,34-38]. As mentioned before, heavy oils are laden with asphaltenes in the bulk which imparts them with their semi-solid structure. Breaking or more technically called ‘cracking’ of the asphaltene structure is the first step in making the oil more accessible for further treatment [35-37]. Catalytic cracking also distributes the asphaltene aromatic structure into lighter fractions which increases the value of the oil. Involving nanotechnology in the field of heavy oil recovery is a way of exploring efficient ways to implement the same process but with improved results [39].

Although so far, many nanomaterials have been developed to improve HO recovery, there is still work to be done to improve the quality of products obtained during the cracking of heavy oil fractions. Well-designed nanoparticles can achieve this goal, which should present a high affinity for heavy oil fractions (asphaltenes), that subsequently can be decomposed into lower molecular weight hydrocarbons and high calorific gases (like hydrogen and others) by the interactions between steam and the catalytic active sites of nanoparticles [19,27,40-49].”

Also, the following paragraph was organized to highlight the study of hydrogen fugacity as one of the main objectives of this study. Paragraph

“In this context, this study looks for an alternative to implement the energy transition strategies. It is well known that renewable energy sources should incorporate traditional energy sources to be more sustainable [7]. Hence, the application of tailor-made nanofluids for the revaluation and production of HO and EHO, in parallel, will entail obtaining H2 as a transitory and complementary source of energy that will help the implementation of this fuel on a large scale until it is achieved the development of 100% “eco” technologies that allow a sufficient supply of green H2. However, the properties of hydrogen, such as the small size of the molecule, make it have a great transport capacity in a porous medium, even with almost impermeable properties [50]. Thereby, it is imperative to analyze the thermodynamic characteristics of the H2 produced in the reservoir during the implementation of nanotechnology-assisted steam injection.”

“….Next, a thermodynamic analysis of the fugacity of hydrogen was done to have a clearer landscape of its in-situ behavior. Based on this analysis it was possible to determine the tendency of hydrogen to be trapped in the reservoir and its dissipation into the porous media.”

Issue 2: The structural characterization of inorganic nanoparticle catalyst is too simple. Usually, XRD, XPS and Raman spectroscopy are used to analyze the crystal structure of inorganic particles. In addition, how to doping the original Al2O3 and CeO2 with Ni and Pd needs to be explained. How can the doping be proved to be successful?

Response: We understand the reviewer’s concern and we agree with it. This work compiles the basic characteristics of both nanoparticles like surface area, hydrodynamic diameter, and crystal size and dispersion because of the robust characterization was published on doi.org/10.3390/nano11051301 for CeO2 and doi.org/10.3390/pr9061009 for Al2O3nanoparticles. On the other hand, the doped amount of Ni and Pd was ensured by Energy Dispersive X-ray spectroscopy (EDX) using a Field Electron and Ion (FEI) microscope model Quanta 400 (SEM) (Eindhoven, The Netherlands) coupled to the EDX source. The following phrases were added to clear both issues:

“Nickel and palladium were doped through the incipient wetness technique [63]. The doped amount of Ni and Pd was ensured by Energy Dispersive X-ray spectroscopy (EDX) using a Field Electron and Ion (FEI) microscope model Quanta 400 (SEM) (Eindhoven, The Netherlands) coupled to the EDX source.

Both materials have been extensively characterized and reported by Cardona et al. [63] and Medina et al. [64].”

Issue 3: The author only analyzed the gas products after the cracking of two kinds of crude oil, but did not study the liquid products in depth. 

Response: The reviewer mentioned that the liquid products were not characterized in depth. Based on that, section 2.3 Effluent analysis was improved. We contemplate API gravity, oil viscosity, and SARA distribution for the analysis.

“Panels a and b of Figure 2 show the API gravity values for untreated crude oil, crude oil after steam injection, crude oil recovered by nanofluid injection dispersed in the steam stream and after a soaking time of 12 h for HO1 and HO2, respectively. For HO1, the API gravity increased after steam injection without the nanoparticles from 12.4° to 12.6°. The results showed increases in the API up to 18° and 18.5° before and after the soaking treatment (panel a). Similarly, for HO2, API gravity remained constant after steam injection without NF (Panel b). Then, during the nanofluid injection, the API gravity increased to 29° and after the soaking stage to 29.2°. To understand the difference in API gravity changes, it is essential to analyze the compositional changes in the content of the SARA fractions. These results are shown in panels c and d for HO1 and HO2, respectively. No appreciable or significant change in asphaltene content is observed during the first steam injection. The distribution of the rest of the SAR components is similar in both crude oils. This result agrees well with the unchanged API values described above. Likewise, several works in the literature report that steam does not modify the chemical composition of crude oil [15,30].

 The presence of the NF generates a decrease from 1.76% to 0.7% and 0.2% of the asphaltene content in the HO1 for the respective stage after steam injection with nanofluid dispersed in its stream and subsequent steam injection after the 12 h soaking of the nanofluid with the porous medium, demonstrating the high catalytic activity of the catalyst. In the case of the HO2, the asphaltene content was reduced from 2.01% to 0.3% and 0.1% in a mass fraction in the same stages. Both crude oils had a very low final asphaltene content. However, the saturates and resins distribution change to a greater extent between both samples. In the HO1, saturates increased from 19% to 26% in mass fraction, whereas in HO2, they increased from 17% to 46% in mass fraction. Finally, resin content was reduced by around 22% (untreated crude oil) and 50% (crude oil recovered after soaking of NF and steam injection) for HO1 and HO2, respectively. According to these results, the injected NP1 attack both asphaltenes and high molecular weight resins, increasing the content of lighter hydrocarbons such as saturates. In this way, better quality is obtained in the HO2.

Many factors can explain the good performance of NP1. First, the combined selectivities and reactivities of Ni and Pd toward the asphaltene and resin molecules result in a decrease in the content of both fractions. Also, because of strong metal support interactions (SMSI) alumina nanoparticles avoid metal sintering after the doping process, which leads to an increase in the number of active sites available for gasification reactions.  Finally, the species –O and –OH resulting from the dissociative adsorption of steam by the alumina lower valence state can be transferred to nickel and palladium and react with surface carbonaceous species [46]. Besides, through the movement of oxygen vacancies formed by the change in the oxidation state of the alumina species and the destabilization of the same, the reagents are transferred to the active sites of the transition element oxides.

Finally, panels e and f depict the viscosity values at a shear rate of 10 s-1 and 25°C of the HO1 and HO2, respectively. During the first stage, oil viscosity was slightly reduced for both samples due to the reduction of cohesive intermolecular forces between asphaltenes and resins.

For the nanotechnology-assisted scenarios (2nd and 3rd stage), a significant reduction in oil viscosity was noted, being higher for the effluent recovered after 12 h of soaking. The oil viscosity for HO1 and HO2 recovered in the third stage was 300 and 104 cP, respectively. The main mechanism that explains the reduction in oil viscosity is the cracking/redistribution of asphaltene-resin systems and their subsequent stabilization through free radical hydrogenation to prevent the formation of heavier compounds.

Interestingly, crude oil upgrading was more noticeable in HO2 than in HO1. The catalytic activity of the material promoted higher API values and lower viscosities in the HO2 sample, probably due to interactions with its heaviest fractions. The chemical nature of the asphaltenes and resins of each crude oil greatly influences the response in gasification reactions for oil upgrading. In this sense, it is to be expected that the HO2 fractions are energetically easier to transform into lighter compounds.”

Issue 4: The authors have not to study the catalytic mechanism of the catalyst, such as what is the catalytic center? What is the relationship between catalytic efficiency and catalyst structure? What crude oil components can be cracked by the catalyst? What is the cracking mechanism of these components?

Response: The reviewer is raising an interesting point here. We agree with the importance of emphasizing the reaction mechanism that allows crude oil upgrading. In previous studies, we have explained how nanoparticles can transform the heavies oil fractions into lower molecular weight hydrocarbons and gases. Based on that, we include some of the most relevant information in the discussion of the manuscript as shown below:

“Many factors can explain the good performance of NP1. First, the combined selectivities and reactivities of Ni and Pd toward the asphaltene and resin molecules result in a decrease in the content of both fractions. Also, because of strong metal support interactions (SMSI) alumina nanoparticles avoid metal sintering after the doping process, which leads to an increase in the number of active sites available for gasification reactions.  Finally, the species –O and –OH resulting from the dissociative adsorption of steam by the alumina lower valence state can be transferred to nickel and palladium and react with surface carbonaceous species [46]. Besides, through the movement of oxygen vacancies formed by the change in the oxidation state of the alumina species and the destabilization of the same, the reagents are transferred to the active sites of the transition element oxides.”

“….The benefit of nanoparticles in heavy oil gasification is elucidated when hydrogen production is increased, and CO2 release is reduced after the catalytic process. Hydrogen is a typical product of free radical reactions of the heavier molecules in crude oil. Therefore, the nanoparticles are expected to promote these reactions under the conditions evaluated, increasing the amount produced. A previous study demonstrated that hydrogen could be produced from the ssteam-catalyticgasification of asphaltenes and resins [34]. First, the reaction between H2O and C atoms in both fractions releases hydrogen as a direct byproduct [52]. Other reactions, including water-gas shift and steam reforming, were also evidenced by the authors [52]. Nanoparticles could facilitate the production of H2 from o H2O-CH4 and H2O-CO through steam reforming and water-gas shift reactions, respectively [53,54].

Moreover, the presence of Ni and Pd phases and their interactions with Al2O3 support benefit the production of H2. For example, H2 can be produced by Ni/Al2O3 phases by the complete combustion of CH4, H2O, and CO reforming. On the other hand, the Pd/Al2O3 can simultaneously produce different species like H2 and CO. Both systems follow different reaction pathways because of the further transfer of electrons between the active phase to the support [55,56].”

Issue 5: It can be seen that the resin content of both crude oils is much higher than that of asphaltene. Is the main object of catalytic cracking resins? What is the reaction equation for its cracking?

Response: The reviewer is making a good point here. We included the results of resin adsorption and conversion to clarify the effect of the nanoparticles on crude oil upgrading. Briefly, we obtained that both nanoparticles are highly selective for resins and can decompose them at temperatures below 250 °C. The nanoparticles work well for adsorbing and decomposing asphaltenes and resins. These results explain the favorable results obtained on the dynamic tests in which asphaltene and resin content is reduced after nanofluid injection. Below are shown the results:

“….On the other hand, Figure S4 shows resin adsorption isotherms for both crude oils (panels a and b). The nanoparticles exhibit a type Ib adsorption isotherm for resins adsorption. For HO1 and HO2, NP1 uptake was higher than NP2. The difference in resins adsorption between NP1 and NP2 is around 0.23 (resins from HO1) and 0.25 mg m-2(resins from HO2), respectively. Compared with asphaltene adsorption, nanoparticles adsorb a similar amount of resins, which indicates a good selectivity for both heavy compounds. Figure S5 shows the conversion of resins evaluated at isothermal conditions. The profiles show that resins conversion achieves 100% when nanoparticles catalyze the reaction, otherwise, just 30% of resins can be converted at the evaluated conditions. The time to decompose 100% of adsorbed resins increases in the order NP2 < NP1 for both samples, following the same trend of asphaltenes. All these results highlight the NP1 capacity to absorb and decompose asphaltene and resins over NP2.”

a)

b)

Figure S4 Experimental adsorption of resins isolated from a) HO1 and b) HO2 over NP1 and NP2 (dotted lines represent the SLE fitting).

a)

b)

Figure S5. Isothermal conversion for steam gasification of resins isolated form a)HO1 and b)HO2 with and without NP1 and NP2.  Heating rate: 10 °C·min-1, N2 flow: 100 ml·min-1, H2O(g) flow: 6.7 ml·min-1and sample mass 1 mg.

About the reaction equation, we are working on molecular dynamic simulation to describe how to lie the cracking and catalytic cracking of asphaltenes and resins is done based on the reactive force field (REAX-FF). Here we show a scheme to illustrate that. Initially, the breaking of S=O bond occurs. Then the breaking of COOH occurs, followed by the breaking of C-H chain. At higher temperatures the breaking of the aromatic core and aromatic sulfur takes place. We hope to share the detailed results in a future work.

Figure 1. Reaction pathway of asphaltene.

Issue 6: Although fugacity calculation has been used to study the effect of hydrogen, the experimental data support is still not strong enough.

Response: We thank the reviewer for his/her valuable comment. We are aware that an explanation of why analyzing hydrogen fugacity in porous media was missing. Our main objective is to elucidate whether, under the presence of nanocatalysts, the tendency of the underground-produced hydrogen is to be trapped in the reservoir or will dissipate into the porous media. The following was added to the manuscript:

In this context, this study looks for an alternative to implement the energy transition strategies. It is well known that renewable energy sources should incorporate traditional energy sources to be more sustainable [7]. Hence, the application of tailor-made nanofluids for the revaluation and production of HO and EHO, in parallel, will entail obtaining H2 as a transitory and complementary source of energy that will help the implementation of this fuel on a large scale until it is achieved the development of 100% “eco” technologies that allow a sufficient supply of green H2. However, the particular properties of hydrogen, such as the small size of the molecule, make it have a great transport capacity in a porous medium, even with almost impermeable properties [50]. Thereby, it is imperative to analyze the thermodynamic characteristics of the H2 produced in the reservoir during the implementation of nanotechnology-assisted steam injection.”

Also, with the chemical composition of the released gas mixture during the thermal process, it is possible to obtain the fugacity of each specie. Its analysis can be done through cubic or virial equations of state in a mixture of gases or through correlations obtained from experimental information. For this research, the fugacity of the gas mixture released during the displacement test of HO1 and HO2 by steam injection assisted by nanotechnology was calculated. The gas mixture contains hydrogen (H2), ethylene (C2H4), methane (CH4), carbon dioxide (CO2), carbon monoxide (CO), and hydrogen sulfide (H2S).

The application of virial equations of state truncated in the second term with mixing rules was considered to represent the possible intermolecular interactions between pairs of components. The truncation to the second term is done because the interaction between pairs of molecules in gases is more likely than between triples or higher-order interactions.

The mixing rules allow calculating properties for pseudo components that represent the interaction between pairs of molecules. The mixing properties considered were acentric factor , critical temperature , critical compressibility factor , critical molar volume , and critical pressure . The mixing rules are applied to the critical conditions because they are characteristic of each compound and allow the evaluation of the deviation from ideality from the concept of reduced property, which is explained by the theorem of corresponding states.

Reviewer 2 Report

This manuscript describes the in-situ heavy oil upgrading using nanofluid catayst. I recommend this manuscript be accepted for this journal after revising the following points. There are some parts that are a little difficult to understand, so I hope that they can be corrected.

1. Line 171-172, what is the remaining 9%? In what form and where does it exist?

2. In Figure 1, what does VPWE Mean?

3. In Figure 1, did each stage occur sequentially? What does pressure drop mean?

4. Line 216-217, are the HO1 and HO2 results compared to the same stage?

5. Stage 3 had a soaking time of 12 hours. Are stages 1 and 2 comparable in response time? Isn't the better stage 3 result due to the longer soaking time? Or, does it mean that there is no need for soaking since stages 2 and 3 are similar?

6. In Figure 2 and 3, please check that the alphabet of the figure caption matches the graph.

7. If hydrogen is generated, it will be mainly used for heavy oil upgrading. Detailed explanation of what the fugacity calculation means seems to be necessary.

8. In Figure 6 and 7, please provide detailed descriptions of a and b in the caption.

Round 2

Reviewer 1 Report

The authors have carefully revised the manuscript following the comments of reviewers. Now it can be accepted.

Reviewer 2 Report

The authors tried to answer the reviewer’s questions. The answers were helpful in better understanding. I recommend this manuscript be accepted for this journal.